# Prevalence and predictors of clinically significant health anxiety among Saudi medical students: A multi-university cross-sectional study

Mohamed Terra[1,☯,*], Mohamed Baklola[2,☯,*], Naji Al-bawah[☉,3,*], Abdullah Hassan Alhalafi[4], Abdullah M. Alshahrani[4], Fatimah Ahmed Alhammad[5], Naif Hameed Aljohani[6], Sulaiman Mohammed A. Alahmad[☉,7], Miad Ali Alhamrani[8], Yara Musleh Alsolami[8], Reema Nasser Alshaya[9], Najim Z. Alshahrani[10,*]

**1** General practitioner, Ministry of Health, Cairo, Egypt, **2** Intern doctor, Mansoura University Hospital, Mansoura, Egypt, **3** Faculty of Medicine, Sana#39;a University, Sana#39;a, Yemen, **4** Department of Family and Community Medicine, College of Medicine, University of Bisha, Bisha, Saudi Arabia, **5** Family Medicine Resident, Alahsa Family Medicine Academy, Alahsa, Saudi Arabia, **6** Family Medicine Resident, Makkah Health Cluster, Ministry of Health, Jeddah, Saudi Arabia, **7** SBPM Resident, Saudi Board of Preventive Medicine, Saudi Arabia, **8** College of Medicine, University of Jeddah, Jeddah, Saudi Arabia, **9** IBN Sina National College, Jeddah, Saudi Arabia, **10** Department of Family and Community Medicine, Faculty of Medicine, University of Jeddah, Jeddah, Saudi Arabia

☯ These authors contributed equally to this work.
* Najialbawah@gmail.com (NAB), Mohamedbaklola2000@gmail.com (MB), Mohamedtera75@gmail.com (MT), nalshahrani@uj.edu.sa (NZA)

## Abstract

### Background

Health anxiety is a common yet underrecognized concern among medical students, who are routinely exposed to illness-related content and academic pressure. This study aimed to estimate the prevalence of clinically significant health anxiety among Saudi medical students and identify its predictors. Secondary objectives were to examine its association with quality of life and explore disease-related fears.

### Methods

A descriptive, cross-sectional study was conducted across multiple Saudi universities between November 2024 and April 2025. Participants were recruited via convenience sampling, and data were collected using a structured online questionnaire. The instrument included sociodemographic items, the Short Health Anxiety Inventory (SHAI), a self-perception of illness question, and the SF-12 to assess quality of life. Descriptive statistics, Chi-square tests, Mann–Whitney U tests, Pearson correlation, and logistic regression analysis were used to analyze the data.

### Results

A total of 650 students completed the survey. Clinically significant health anxiety was reported in 30% of participants. Health anxiety prevalence was higher among

**Data availability statement:** The minimal underlying dataset supporting the findings of this study is publicly available in the Figshare repository at https://doi.org/10.6084/m9.figshare.30572492.v1.

**Funding:** The author(s) received no specific funding for this work.

**Competing interests:** The authors have declared that no competing interests exist.

females compared to males (36.3% vs. 25.8%), as well as among students in their second academic year, rural residents, private university students, and those with personal or family psychiatric history. Logistic regression identified several significant predictors: female students demonstrated 2.3-fold higher odds of health anxiety compared to males (AOR = 2.33, 95% CI: 1.51–3.62), rural residence, private university enrollment (AOR = 6.14, 95% CI: 3.66–10.5), and psychiatric history. A significant negative correlation was found between health anxiety and quality of life (r = −0.25, p < 0.001). Students with health anxiety reported notably lower quality of life scores. Additionally, a network analysis of disease-related fears revealed cancer as the most frequently reported concern, followed by cardiovascular and endocrine conditions. These fears clustered into somatic, chronic, and neuropsychiatric domains.

## Conclusion

Clinically significant health anxiety affects nearly one-third of Saudi medical students and is associated with key sociodemographic and academic risk factors, as well as poorer quality of life. These findings underscore the need for mental health support systems and preventive strategies within medical education environments to address and manage health-related anxieties early.

## Background

Anxiety disorders represent the most prevalent category of psychiatric illnesses globally, with an estimated prevalence of 7.3% and a wide range between 4.8% and 10.9% [1]. Women are disproportionately affected, with adolescent females being up to twice as likely as males to experience anxiety-related conditions [2]. According to the World Health Organization, anxiety disorders ranked among the top six causes of years lived with disability (YLDs) worldwide in 2015 and were even more burdensome in high-income countries [3].

A particular subtype, known as health anxiety (HA), involves persistent worry about having a serious illness, often triggered by normal bodily sensations or minor symptoms [4]. The estimated lifetime prevalence of health anxiety is around 5.7%, with a current prevalence of 3.4%. This form of anxiety is especially relevant among medical students, a population exposed to extensive health-related information and high academic pressure [5]. Commonly referred to as "medical student syndrome" or nosophobia, HA among this group often manifests through exaggerated concern about self-diagnosed illnesses, which may shift based on recent academic or clinical exposure [6].

Health anxiety and hypochondriasis are sometimes used interchangeably, though HA is often considered a milder form, characterized more by anxiety and less by delusional or depressive features [6]. The proliferation of online health resources and symptom-checking tools has further exacerbated HA, with "cyberchondria" now recognized as a factor contributing to its persistence and severity [7].

Several international studies have examined HA among medical students. Research from India reported that nearly one in seven students experienced health anxiety, with preclinical students being the most affected [8]. In the United Arab Emirates, nearly 10% of medical students were found to have HA, often linked to previous experiences with physical or mental illness [9]. In Egypt, a study conducted across ten medical schools revealed that 15.7% of participants experienced clinically significant health anxiety [6]. The prevalence was notably higher among female students (17.5%) and those who reported dissatisfaction with their academic performance (18%) [6]. Further studies noted that while medical students generally demonstrate higher mental health literacy than their non-medical peers, many distressed students still avoid seeking help due to self-reliance and stigma [10,11].

In Saudi Arabia, the prevalence of illness anxiety among adults in Taif was 25.3%, with younger age, female gender, single marital status, and prior hospitalization for cardiac symptoms being significant risk factors [12]. Among Saudi medical students, a recent study identified illness anxiety disorder in 17% of participants, particularly in younger students and those with recent medical visits [13]. National data from 2022 revealed that 12.4% of Saudis were at risk of generalized anxiety disorder, although treatment rates remained low [14]. Although health anxiety may negatively affect student well-being, many studies in this area primarily focus on estimating prevalence and identifying associated factors, with fewer studies assessing quality-of-life outcomes in a structured way.

Therefore, this study aimed to estimate the prevalence of clinically significant health anxiety among Saudi medical students and to identify its sociodemographic and academic predictors across multiple universities. As secondary objectives, we assessed the association between health anxiety and quality of life and explored patterns of disease-related fears reported by students.

## Methods

### Study design and study period

This descriptive, cross-sectional study with an analytical component was conducted over six months, from November 2024 to April 2025. The research included multiple medical universities across different regions in Saudi Arabia, allowing for diverse student representation.

### Sample size calculation

The sample size was estimated using the OpenEpi online calculator [15], with the primary variable being the prevalence of health anxiety among medical students. Drawing on previous findings from a study involving Egyptian university students, the anticipated prevalence was set at 15.7% [6]. With a confidence level of 95%, statistical power of 80%, and a 5% margin of error, the minimum required number of participants was calculated to be 202. To improve subgroup analyses and compensate for potential biases, the sample size was adjusted by a design effect of 3, bringing the required total to 606. A final count of 650 students completed the questionnaire.

### Sampling strategy and data collection

Participants were recruited through a convenience sampling method. The survey link was distributed to medical students across eight universities in Saudi Arabia (five public and three private). Based on available enrollment data, the estimated total population of medical students across these institutions was approximately 12,000–13,000 students. The survey was circulated through official student email lists, university social media groups, and student organization networks, with an estimated potential reach of around 8,000 students. Because the survey relied on online dissemination and voluntary participation, an exact response rate could not be calculated; however, the distribution strategy ensured broad national representation across academic years and institution types.

The survey was hosted on Google Forms, ensuring anonymous and flexible participation. All medical students, regardless of academic year or specialization, were eligible, which allowed for a diverse representation in terms of gender and field of study.

## Survey instrument and measures

Data were collected through a structured, self-administered online questionnaire consisting of four main sections. The first section addressed sociodemographic characteristics, including age, gender, place of residence, academic performance, psychiatric history, and whether participants experienced external pressure to pursue a medical degree. Past psychiatric history was assessed using a mandatory binary (yes/no) question capturing any prior psychological or psychiatric condition. In contrast, participants were also presented with an optional multiple-response question to specify the type of condition (e.g., depression, anxiety, sleep disorders). Because this item was not mandatory and allowed multiple selections, not all participants who reported a psychiatric history provided details on specific diagnoses.

The second section utilized the Short Health Anxiety Inventory (SHAI), a validated and widely used tool for assessing health anxiety [16]. This version included 18 items divided into two subscales: one evaluating general health anxiety and the other assessing illness-related distress. Each item was rated on a scale from 0 to 3, with a total possible score ranging from 0 to 54. A score of 27 or higher was used to indicate clinically significant levels of health anxiety [16]. The third section consisted of a single question asking participants to identify any diseases they believed they might have, aiming to capture their subjective perceptions of health.

The final section evaluated quality of life using the second version of the 12-item Short Form Survey (SF-12), developed by Ware, Kosinski, and Keller [17]. This instrument measures eight domains reflecting both physical and mental aspects of well-being, including general health, physical functioning, role limitations due to physical and emotional problems, bodily pain, vitality, mental health, and social functioning. Although the SF-12 is often scored using norm-based algorithms to generate Physical (PCS-12) and Mental (MCS-12) Component Summary scores, in the present study we used the SF-12 as a single overall QoL composite score for within-sample comparisons. This approach has been previously applied in studies examining health anxiety and quality of life among medical students using the SHAI and SF-12 together [6]. Item responses were coded so that higher values reflected better perceived health-related quality of life, and a total composite QoL score was computed by summing the coded responses across SF-12 items. The resulting score was treated as a continuous non-normed QoL indicator and summarized using median (IQR).

## Statistical analysis

Data analysis was performed using SPSS version 27 for macOS. Continuous variables were expressed as means with standard deviations or as medians when not normally distributed. Categorical data were presented as frequencies and percentages. Associations between categorical variables were evaluated using the Chi-square test, while differences in non-normally distributed continuous variables were assessed using the Mann–Whitney U test. Pearson's correlation was used to explore relationships between health anxiety and quality of life scores. Variables showing significance in univariate analysis were entered into a binary logistic regression model to identify independent predictors of health anxiety. Statistical significance was defined as a p-value of 0.05 or less. To assess multicollinearity among predictors, we examined variance inflation factors (VIFs) and tolerance statistics prior to model interpretation. VIF values > 5 were considered indicative of problematic collinearity.

We conducted a network analysis to explore the co-occurrence of disease-related fears among respondents. Each disease category was represented as a node, and edges between nodes reflected the frequency of co-reporting two fears within the same participant.

The network was constructed using the Fruchterman–Reingold force-directed algorithm in the qgraph package (R). This layout arranges nodes in two-dimensional space such that more strongly connected nodes are placed closer

together. Edge thickness is proportional to the pairwise co-occurrence frequency, while node size represents betweenness centrality. Betweenness centrality was calculated as the proportion of shortest paths in the network that pass through a given node, indicating its importance as a "bridge" between different disease fears.

Given the exploratory nature of this study, no formal correction for multiple comparisons was applied. We relied on effect sizes, confidence intervals, and consistency with prior literature to guide interpretation.

### Ethical approval and consent to participate

This study was approved by the University of Jeddah Research Ethics Committee (UJ-REC) (Registration No. HAP-07-J-073, UJ-REC-292) and conducted in accordance with the Declaration of Helsinki. Participation was voluntary. Informed consent was obtained electronically from all participants prior to survey initiation: before accessing the questionnaire, participants were required to read an online consent form detailing the study objectives, procedures, risks, and confidentiality measures, and then indicate agreement by selecting a "Yes, I agree to participate" option. Only those who provided consent were able to proceed to the survey. No personal identification information was collected. All participants were aged 18 years or older; therefore, no parental or guardian consent was required. The ethics committee did not waive any consent requirements.

## Results

### Demographic characteristics and health anxiety prevalence

The study included 650 participants, comprising 455 (70%) without health anxiety and 195 (30%) with health anxiety (Table 1). Age distributions differed significantly between groups (median 23 years [IQR 22–24] vs. 23 years [IQR 21–25], p=0.04). Academic level showed marked variation (p<0.001), with Year 2 students having nearly equal proportions of health anxiety (48.5%) and no anxiety (51.5%), while Year 6 students had the lowest prevalence (5.4%). Gender differences were significant (p=0.004), with 36.3% of females and 25.8% of males experiencing health anxiety. Rural residents had a higher prevalence (43.1% vs. 25.7%, p<0.001), as did private university students (67.3% vs. 22.9%, p<0.001). Academic satisfaction was not statistically significant (p=0.081). Having a family member in the medical field also showed no association (p=0.9). However, participants with a family psychiatric history (36.1% vs. 25.8%, p=0.005) or a personal psychiatric history (37.8% vs. 62.2%, p<0.001) reported significantly higher health anxiety prevalence.

### Demographic characteristics and Quality of Life assessment

Quality of life (QoL) scores demonstrated significant variation across academic levels (p<0.001), with Year 1 students reporting highest QoL (median 31 [IQR 28−32]) and Year 5 students the lowest (27 [24 -29]) (Table 2). Academic satisfaction showed strong association with QoL (satisfied: 28 [26-32] vs unsatisfied: 26 [24-29], p<0.001). Participants without psychiatric history reported significantly better QoL (29.5 [27-33]) than those with history (27 [25-30], p<0.001). Family medical background showed modest QoL association (with medical family: 28 [26-31] vs without: 27 [25-31], p=0.020). QoL values represent the overall SF-12 composite score (non–norm-based) used for internal comparisons and are not directly comparable to norm-based PCS-12/MCS-12 values.

### Multivariable analysis of health anxiety predictors

The logistic regression model revealed several significant predictors (Table 3). Academic year showed nonlinear effects, with Year 2 students having 3.25-fold increased odds (95% CI: 1.53–7.08, p=0.002) and Year 6 students showing 71.8% reduced odds (AOR=0.282, 95% CI: 0.0779–0.891, p=0.031) of health anxiety compared to Year 1. All academic years were included in the regression model; however, Year 4 did not show a statistically significant association and is not presented in Table 3. Male gender demonstrated protective effects (AOR=0.430, 95% CI: 0.276–0.664, p<0.001), as did

**Table 1. Demographic characteristics and prevalence of health anxiety among participants.**

| Characteristic | Health anxiety | | Total (N) | P-value² |
|---|---|---|---|---|
| | **No** | **Yes** | | |
| | **N = 455 (Row %)** | **N = 195 (Row %)** | | |
| **Age¹** | 23 (22–24) | 23 (21–25) | 23 (21-25) | **0.04*** |
| **Academic level** | | | | **<0.001*** |
| Year 1 | 54 (72.0%) | 21 (28.0%) | 75 | |
| Year 2 | 50 (51.5%) | 47 (48.5%) | 97 | |
| Year 3 | 43 (55.8%) | 34 (44.2%) | 77 | |
| Year 4 | 82 (70.7%) | 34 (29.3%) | 116 | |
| Year 5 | 139 (72.0%) | 54 (28.0%) | 193 | |
| Year 6 | 87 (94.6%) | 5 (5.4%) | 92 | |
| **Gender** | | | | **0.004*** |
| Female | 167 (63.7%) | 95 (36.3%) | 262 | |
| Male | 288 (74.2%) | 100 (25.8%) | 388 | |
| **Residence** | | | | **<0.001*** |
| Rural | 91 (56.9%) | 69 (43.1%) | 160 | |
| Urban | 364 (74.3%) | 126 (25.7%) | 490 | |
| **University type** | | | | **<0.001*** |
| Governmental | 421 (77.1%) | 125 (22.9%) | 546 | |
| Private | 34 (32.7%) | 70 (67.3%) | 104 | |
| **Satisfied with academic performance** | | | | 0.081 |
| No | 172 (66.2%) | 88 (33.8%) | 260 | |
| Yes | 283 (72.6%) | 107 (27.4%) | 390 | |
| **Family member in medical field** | | | | 0.9 |
| No | 201 (70.0%) | 86 (30.0%) | 287 | |
| Yes | 254 (69.9%) | 109 (30.1%) | 363 | |
| **Family history of psychiatric disorder** | | | | **0.005*** |
| No | 287 (74.2%) | 100 (25.8%) | 387 | |
| Yes | 168 (63.9%) | 95 (36.1%) | 263 | |
| **Past psychiatric disorder** | 280 (62.2%) | 170 (37.8%) | 450 | **<0.001*** |

¹ Median (Q1, Q3); n (%), ² Wilcoxon rank sum test; Pearson's Chi-squared test. * Significant level at p value <0.05

urban residence (AOR = 0.433, 95% CI: 0.272–0.689, p < 0.001). Strikingly, private university attendance conferred 6.14-fold increased odds (95% CI: 3.66–10.5, p < 0.001). Psychiatric history showed strong associations, with family history doubling odds (AOR = 2.11, 95% CI: 1.39–3.21, p < 0.001) and personal history nearly quadrupling odds (AOR = 3.74, 95% CI: 2.27–6.36, p < 0.001).

Model fit indicators suggested that the logistic regression demonstrated adequate fit and good discriminatory power (Hosmer–Lemeshow test: $\chi^2$ = 6.12, df = 8, p = 0.63; Nagelkerke $R^2$ = 0.27; AUC = 0.81, 95% CI 0.76–0.86). No evidence of problematic multicollinearity was detected among predictors (all VIF values < 2.5), suggesting that the regression coefficients were stable and interpretable.

## Association between health anxiety and quality of life

A significant negative correlation was observed between health anxiety and quality of life (QoL) scores (r = −0.25, p < 0.001) (Fig 1). As health anxiety levels increased, QoL scores systematically decreased, indicating that higher

**Table 2. Quality of life scores across demographic characteristics of participants.**

| Characteristic | | Quality of Life[1] | P-value[2] |
|---|---|---|---|
| **Gender** | Female | 28 (26 −31) | 0.43 |
| | Male | 27 (25-30) | |
| **Academic Level** | Year 1 | 31 (28-32) | **<0.001\*** |
| | Year 2 | 28 (26-30) | |
| | Year 3 | 28 (25-30) | |
| | Year 4 | 28.5 (26-32) | |
| | Year 5 | 27 (24-29) | |
| | Year 6 | 27 (25-32) | |
| **Residence** | Rural (Countryside) | 28 (25-30) | 0.24 |
| | Urban (City) | 28 (25-31) | |
| **University Type** | Governmental | 28 (25-31) | 0.32 |
| | Private | 28 (25-30) | |
| **Academic Satisfaction** | Not Satisfied | 26 (24-29) | **<0.001\*** |
| | Satisfied | 28 (26-32) | |
| **Family in Medical Field** | No | 27 (25-31) | **0.020\*** |
| | Yes | 28 (26-31) | |
| **Family Psychiatric History** | No | 28 (25-31) | 0.2 |
| | Yes | 28 (26-31) | |
| **Past Psychiatric History** | No | 29.5 (27-33) | **<0.001\*** |
| | Yes | 27 (25-30) | |

[1] Median (Q1, Q3); [2] Wilcoxon rank sum test, Kruskal-Wallis test.* Significant level at p value < 0.05

**Table 3. Multivariable logistic regression analysis of factors associated with Health Anxiety status.**

| Variable | AOR | CI (95%) | P-value |
|---|---|---|---|
| (Intercept) | 0.03 | 0.003 - 0.400 | **0.007\*** |
| **Age** | 1.06 | 0.951 - 1.19 | 0.282 |
| **Academic level:** Year 2 (vs Year 1) | 3.25 | 1.53 - 7.08 | **0.002\*** |
| **Academic level:** Year 3 (vs Year 1) | 2.71 | 1.23 - 6.08 | **0.014\*** |
| **Academic level:** Year 5 (vs Year 1) | 2.66 | 1.22 - 5.95 | **0.015\*** |
| **Academic level:** Year 6 (vs Year 1) | 0.282 | 0.0779 - 0.891 | **0.031\*** |
| **Gender:** Male (vs Female) | 0.430 | 0.276 - 0.664 | **<0.001\*** |
| **Residence:** Urban (vs Rural) | 0.433 | 0.272 - 0.689 | **<0.001\*** |
| **University type:** Private (vs Public) | 6.14 | 3.66 - 10.5 | **<0.001\*** |
| **Family history of psychiatric disorders** | 2.11 | 1.39 - 3.21 | **<0.001\*** |
| **Past history of psychiatric disorders** | 3.74 | 2.27 - 6.36 | **<0.001\*** |

* Significant level at p value < 0.05. Year 1 was used as the reference category. All academic years were included in the model; Year 4 is not shown as it was not statistically significant.

health-related distress was associated with poorer perceived well-being. This relationship remained robust across the sample, suggesting that health anxiety substantially impacts overall quality of life.

**Prevalence of reported mental health conditions** The overall prevalence of self-reported psychiatric history (69.2%) was derived from a mandatory yes/no item as shown in Table 1. In contrast, the reporting of specific psychiatric conditions

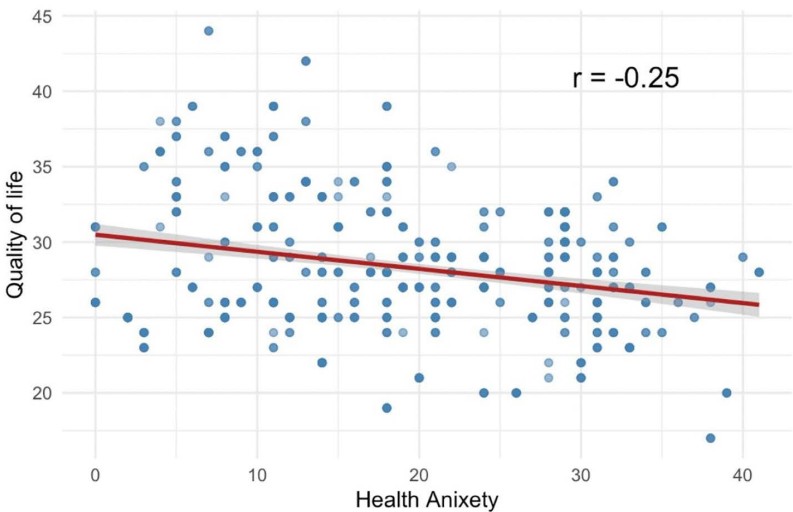

**Fig 1. Scatter plot of health anxiety and quality of life among Saudi undergraduates, highlighting correlation.** The red line represents the fitted linear regression line. A weak negative correlation was observed (r = −0.25, p < 0.001).

(Fig 2) was based on an optional multiple-response question. Consequently, not all participants who reported a psychiatric history selected a specific condition, and the frequencies of individual disorders should be interpreted as partial representations rather than a complete breakdown. Among the study participants, depression was the most frequently reported mental health condition, 16.3%, followed by stress disorders, 12.8% and sleeping disorders, 12.1%. Social anxiety disorder (8.1%) and generalized anxiety disorder (6.4%) were also notable, while less common conditions included OCD (6.1%) and personality disorders (6.0%). The distribution highlights a predominance of mood and stress-related conditions, with depression alone accounting for nearly one-fifth of all reported mental health concerns.

**Disease-related health anxieties and network analysis of disease fear co-occurrence**The study cohort (N = 650) exhibited a distinct hierarchy of disease-related health anxieties. Malignancy concerns were most prevalent, with 40.9% (n = 266) of participants endorsing cancer as a primary fear. Cardiovascular-related health anxieties followed at 29.7% (n = 193), with diabetes mellitus fears closely trailing at 29.1% (n = 189). Comparable prevalence rates emerged for ophthalmic (27.8%, n = 181) and psychiatric (27.7%, n = 180) health concerns. Secondary analyses revealed substantial anxiety regarding neurological (22.9%), reproductive (22.6%), and cardiovascular-metabolic (hypertension: 18.9%) conditions.Network analysis identified three clinically meaningful fear clusters (Fig 3). First, a somatic threat cluster demonstrated strong co-occurrence between oncological and cardiovascular fears, with 32% joint prevalence. Second, a chronic disease cluster showed significant diabetes-hypertension comorbidity (25% co-occurrence). Third, a neuropsychiatric cluster revealed neural-psychiatric fear associations (18%) that significantly exceeded neural-somatic linkages. The network#39;s global architecture exhibited scale-free properties, with cancer fears serving as the principal hub (betweenness centrality = 0.42).

## Discussion

This study investigated the prevalence and predictors of health anxiety (HA) among Saudi medical students and examined its association with quality of life (QoL). Our findings reveal that 30% of the participants experienced clinically significant health anxiety, a prevalence notably higher than that reported in comparable studies from Egypt (15.7%), the United Arab Emirates (9.3%), and Pakistan (11.9%) [6,9,18]. This discrepancy may reflect contextual factors such as differences in university environments, academic demands, sociocultural expectations, and access to mental health support.

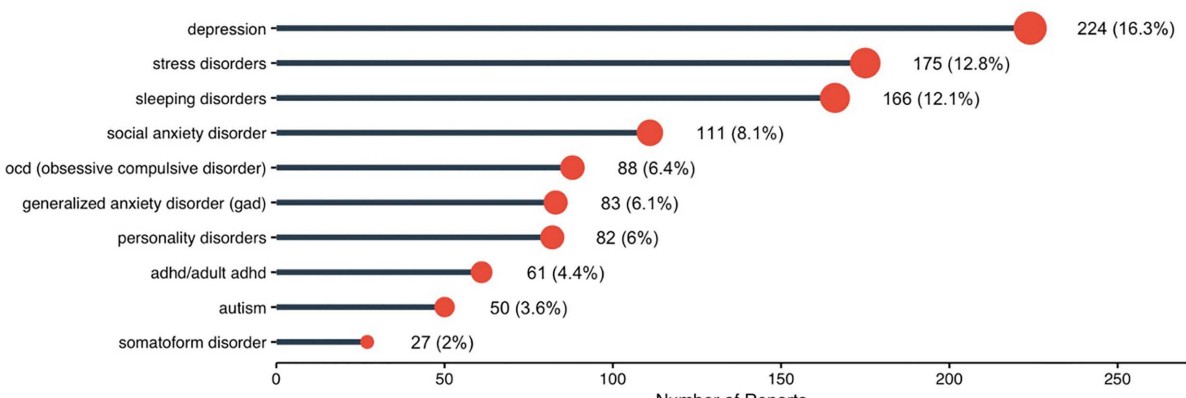

**Fig 2. Common self-reported psychiatric disorders among Saudi undergraduates.** Participants could select multiple conditions, and responses were optional.

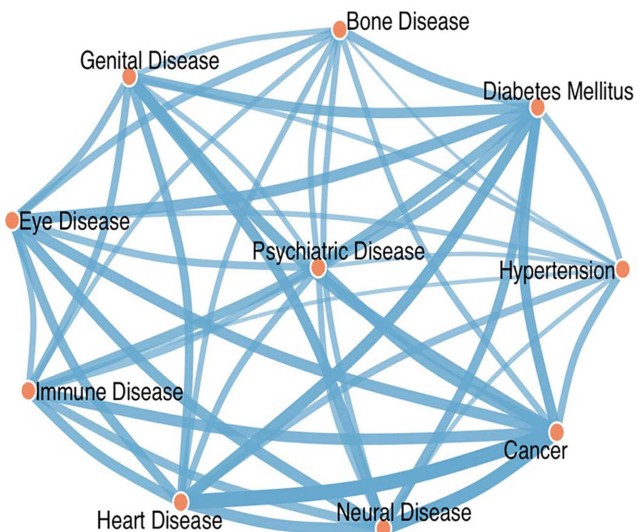

**Fig 3. Network Analysis of disease fear co-occurrence among respondents.** Node size represents betweenness centrality (higher = more connections); edge thickness indicates pairwise co-occurrence frequency.

Consistent with global literature, gender emerged as a significant predictor of health anxiety, with female students reporting substantially higher levels of anxiety than their male counterparts. This trend has been consistently documented in Egypt and Pakistan [6,18], and is often interpreted in light of broader social dynamics. Female students may face unique stressors, including heightened academic expectations, societal pressures, and concerns about future career opportunities, which together may amplify vulnerability to anxiety [19]. Furthermore, epidemiological studies have shown that women are generally more prone to anxiety-related disorders, which may partly explain the pattern observed here [20].

In Saudi Arabia, stigma toward mental health conditions may discourage open reporting of psychological distress, leading some students to frame their anxiety in somatic or health-related terms rather than as a mental health issue [21].

Gender roles and family expectations may also contribute, as female students often face heightened academic and social pressures, in addition to cultural norms that can restrict autonomy and exacerbate internalized worry about health and performance [22,23]. Furthermore, the strong emphasis on family involvement in health decisions may influence patterns of reassurance-seeking and delay independent help-seeking from mental health professionals.

Academic dissatisfaction showed a higher proportion of health anxiety in our sample; however, this association did not reach statistical significance in univariate analysis and was not retained in the multivariable model. Therefore, it cannot be considered an independent predictor in this study. Nevertheless, the observed directional trend is consistent with previous findings from Egypt and other settings [6], suggesting that academic stress may still play a contributory role. The psychological demands of medical education, including continuous exposure to disease-related content and high expectations for academic performance, may heighten students' sensitivity to bodily symptoms and contribute to maladaptive health-related concerns [24,25]. This context may be particularly relevant during the preclinical years, where theoretical instruction often emphasizes pathology without sufficient clinical reassurance. Supporting this interpretation, our multivariable analysis identified second-year students as being at significantly higher risk of health anxiety, a finding that aligns with prior studies from Pakistan and Egypt [6,18].

One of the most striking findings in this study was the markedly higher likelihood of clinically significant health anxiety among students enrolled in private universities (AOR = 6.14). Although our design does not allow causal inference, the magnitude of this association suggests that institutional context may meaningfully shape students' psychological experiences. Importantly, this pattern is consistent with prior Saudi evidence showing that private medical university students report significantly higher levels of depression, anxiety, and stress compared with their counterparts in public universities, indicating that elevated distress in private institutions may extend beyond health anxiety specifically [26].

Several plausible mechanisms may explain this association. First, financial pressure is likely a key contributor. Unlike public universities that are typically state-supported for eligible students, private medical education often involves substantial tuition costs and related expenses, which may increase perceived "stakes" of academic performance and intensify stress-related symptom monitoring [27]. Financial strain may also amplify cognitive preoccupation with health as a feared threat to academic continuity and family investment. Second, private institutions may differ in academic structure and competitiveness, including assessment density, grading pressure, or curriculum pacing, which can elevate stress and reinforce repeated exposure to illness-related content without sufficient psychological buffering [28]. Third, differences in availability and accessibility of student well-being infrastructure, including counseling capacity, confidentiality perceptions, and proactive screening programs, could influence whether students receive early support or continue reassurance-seeking and symptom checking, which are core maintaining behaviors in health anxiety [29].

Beyond structural explanations, cultural and social-contextual factors may also shape how health anxiety emerges and is expressed in different institutional environments. For example, private university students may experience stronger family expectations linked to financial investment, which may increase fear of failure and heighten attention to bodily sensations during periods of stress [30]. Peer norms may also differ: if private-university culture fosters more competitive academic comparison or less open discussion of mental health concerns, students may be more likely to internalize distress and express it through somatic or illness-related worry rather than seeking psychological support [31]. Additionally, religious coping practices and spiritual interpretations of illness may influence health-related fears in nuanced ways, either buffering anxiety through meaning-making and acceptance, or potentially increasing vigilance if illness is perceived as a serious threat requiring constant monitoring [32]. Although these mechanisms were not directly measured in our study, they represent plausible and testable pathways that could help explain why private university enrollment showed such a large association with health anxiety [33].

Future research should therefore adopt mixed-methods approaches to clarify institutional drivers and contextual mechanisms. Quantitative studies could compare private and public institutions using objective indicators such as tuition burden, scholarship availability, academic workload intensity, and student-to-counselor ratios, while qualitative

interviews or focus groups could explore perceived academic pressure, family expectations, stigma, coping strategies, and help-seeking behavior [34]. Such work would guide targeted and institution-specific interventions, particularly within private universities, where preventive mental health screening, early psychoeducation about health anxiety, and accessible confidential counseling may be especially impactful [34].

Personal and family psychiatric history emerged as powerful predictors. Students with a personal history of mental health conditions were nearly four times more likely to experience HA, while those with a family history had double the odds. These findings align with established evidence suggesting that individuals with prior psychological vulnerability or familial predisposition are at increased risk for health-focused anxieties [35].

Importantly, the study also established a significant inverse relationship between health anxiety and quality of life. Students with elevated HA scores consistently reported lower QoL, particularly in mental health domains. This correlation supports findings from previous regional studies and highlights the broader implications of HA for students' well-being [6]. The impact of persistent health concerns can extend beyond psychological distress, affecting sleep, concentration, academic performance, and social engagement [36]. Over time, this can erode resilience and reduce functioning across multiple life domains.

Although the correlation between health anxiety and quality of life was modest, this finding indicates that even relatively small increases in health anxiety can meaningfully interfere with students' daily functioning, academic performance, and well-being. Similarly, the observed odds ratios, while statistically robust, also highlight clinically relevant disparities; for example, male students were substantially less likely to report high health anxiety, suggesting that female students may represent a particularly vulnerable subgroup in need of targeted support [37,38]. Taken together, these results underscore that even small-to-moderate associations can carry important implications for medical education, as they translate into tangible challenges for student support services and mental health policy within universities.

These findings suggest that early interventions, such as resilience training, peer support initiatives, and routine mental health screening, may help mitigate the real-world impact of health anxiety on student wellbeing and academic performance [39,40].

An additional contribution of this study lies in its exploration of students' self-reported disease-related fears. Cancer, cardiovascular disease, and diabetes emerged as the most common sources of health anxiety, forming identifiable clusters of concern. The network analysis further revealed patterns of fear co-occurrence, with oncological concerns serving as a central hub. This may reflect the heightened awareness of these conditions among medical students, potentially exacerbated by their academic exposure to severe clinical cases without balanced reassurance or contextualization [41,42].

Taken together, these findings point to the hidden psychological costs of medical education in Saudi Arabia. Health anxiety appears to be a prevalent and multifaceted issue, shaped by individual, academic, and environmental factors. Its association with diminished quality of life signals the need for early interventions. Medical curricula may benefit from integrating wellness education, resilience training, and accessible psychological services, particularly for students in the early years of training and those with identifiable risk factors [43].

Given the markedly higher odds of health anxiety among students enrolled in private universities (AOR = 6.14), Year 2 students (AOR = 3.25), and female students (AOR = 2.33), institutions should adopt targeted and measurable strategies rather than generalized wellness messaging. First, to reduce tuition-related stress, private universities may implement structured financial protection mechanisms, such as need-based emergency mental health grants and temporary tuition deferral or installment plans for students who screen positive for significant distress, with transparent eligibility criteria (e.g., confirmed high SHAI score and documented functional impairment). Second, medical colleges should strengthen counseling access by adopting minimum service standards, including a minimum counselor-to-student ratio of 1:1,500 with an institutional target of 1:1,000, aligned with commonly referenced university mental health staffing benchmarks

[44]. Institutions may also define service performance indicators such as same-day walk-in triage, guaranteed follow-up appointments within ≤7–14 days, and confidential referral pathways for students with psychiatric history [45].

Third, academic interventions should be risk-stratified: because Year 2 students represented the highest-risk academic group in our model, schools should implement mandatory peer/near-peer mentoring programs during Year 2 [46], pairing junior students with trained senior mentors, and integrating structured sessions addressing coping with symptom-focused anxiety, academic stress, and health-related reassurance-seeking behaviors; peer mentoring models in medical education have shown academic and psychosocial benefits and are feasible within resource-limited settings. Fourth, universities should introduce routine screening protocols using validated instruments such as the SHAI (as applied in this study) during predictable high-stress periods (e.g., midterms and final examination months), with standardized follow-up for high-risk subgroups (females, rural residents, private university students, and those with personal or family psychiatric history) [45]. Finally, to ensure accountability, universities should evaluate these interventions using measurable outcomes such as reductions in SHAI scores over time, counseling utilization rates, wait-time indicators, academic retention, and student-reported quality of life.

## Limitations

This study has several limitations. First, the use of a cross-sectional design prevents causal inferences between health anxiety and associated factors. Second, the reliance on self-reported data may introduce reporting bias or social desirability effects, particularly regarding psychiatric history and academic satisfaction. Third, the recruitment strategy relying on online questionnaires disseminated via social media and university channels may have led to selection bias, as students with a greater interest in health and mental well-being were more likely to participate, potentially resulting in an overestimation of the prevalence of health anxiety. Additionally, while validated instruments were used, cultural nuances may influence how health anxiety and quality of life are perceived and reported. The assessment of specific psychiatric conditions was based on a non-mandatory multiple-response item, which may have led to underreporting of individual disorders. Therefore, the distribution of specific conditions should be interpreted with caution and may not fully reflect the true composition of psychiatric history in the sample. Future studies using longitudinal designs and random sampling are recommended to confirm and extend these findings. Data collection occurred between November and April, overlapping with midterm examinations (November–December), clinical rotation transitions (January), and final exam preparation (March–April). These periods are typically associated with elevated academic stress, which may have contributed to higher observed rates of health anxiety. Future studies should account for academic calendar phases or conduct data collection at multiple time points to better control for this effect.

Because multiple statistical tests were conducted, some associations may have reached significance by chance. While no formal correction for multiple comparisons was applied, given the exploratory nature of the study, results should be interpreted cautiously, particularly those with marginal significance. Given the exploratory nature of this study, no formal correction for multiple comparisons was applied. We relied on effect sizes, confidence intervals, and consistency with prior literature to guide interpretation.

## Conclusion

This study highlights a high prevalence of health anxiety among Saudi medical students, affecting nearly one-third of participants, with significant associations observed between health anxiety and lower quality of life. Key predictors included female gender, rural residence, private university enrollment, and personal or family psychiatric history. These findings underscore the psychological burden faced by medical students and the need for targeted mental health support within academic institutions. Addressing health anxiety early through structured interventions, improved access to counseling services, and wellness programs may help mitigate its negative impact on students' well-being, academic performance, and long-term professional development.

## Acknowledgments

Our heartfelt thanks also go to p-value research hub platform. Their training empowered researchers, starting from beginner levels, to undertake this study and become authors. We are immensely grateful for the support provided by this remarkable organization.

## Author contributions

**Conceptualization:** Naji Al-bawah, Abdullah Hassan Alhalafi, Abdullah M. Alshahrani, Fatimah Ahmed Alhammad, Naif Hameed Aljohani, Miad Ali Alhamrani, Reema Nasser Alshaya, Najim Z. Alshahrani.

**Data curation:** Naji Al-bawah, Abdullah Hassan Alhalafi, Abdullah M. Alshahrani, Fatimah Ahmed Alhammad, Naif Hameed Aljohani, Miad Ali Alhamrani, Yara Musleh Alsolami, Reema Nasser Alshaya, Najim Z. Alshahrani.

**Formal analysis:** Abdullah M. Alshahrani, Reema Nasser Alshaya, Najim Z. Alshahrani.

**Investigation:** Naji Al-bawah, Fatimah Ahmed Alhammad, Yara Musleh Alsolami.

**Methodology:** Mohamed Terra, Mohamed Baklola, Naji Al-bawah, Fatimah Ahmed Alhammad, Yara Musleh Alsolami, Najim Z. Alshahrani.

**Project administration:** Mohamed Terra, Mohamed Baklola, Yara Musleh Alsolami, Najim Z. Alshahrani.

**Resources:** Mohamed Baklola, Najim Z. Alshahrani.

**Software:** Mohamed Terra, Mohamed Baklola.

**Supervision:** Mohamed Terra, Mohamed Baklola, Sulaiman Mohammed A. Alahmad, Najim Z. Alshahrani.

**Validation:** Mohamed Terra, Mohamed Baklola, Naif Hameed Aljohani, Sulaiman Mohammed A. Alahmad.

**Visualization:** Mohamed Baklola, Naif Hameed Aljohani, Sulaiman Mohammed A. Alahmad, Reema Nasser Alshaya.

**Writing – original draft:** Mohamed Baklola, Naif Hameed Aljohani, Sulaiman Mohammed A. Alahmad.

**Writing – review & editing:** Mohamed Terra, Mohamed Baklola, Naif Hameed Aljohani, Sulaiman Mohammed A. Alahmad, Najim Z. Alshahrani.

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
