## [Decision Letter · Decision Letter 0]

24 Sep 2025

Dear Dr. Al-bawah,

Thank you for submitting your manuscript to PLOS ONE. After careful consideration, we feel that it has merit but does not fully meet PLOS ONE’s publication criteria as it currently stands. Therefore, we invite you to submit a revised version of the manuscript that addresses the points raised during the review process.

We look forward to receiving your revised manuscript.

Kind regards,

Ahmed Abdelwahab Ibrahim El-Sayed

Academic Editor

PLOS ONE

Journal Requirements:

2. We note that your Data Availability Statement is currently as follows:

“All relevant data are within the manuscript and its Supporting Information files.”

**Additional Editor Comments**

Dear Authors,

Thank you for your submission to PLOS ONE. I have completed my review of your study. The review panel has provided detailed advice and raised several issues that need to be carefully addressed before we can proceed further with your manuscript.

Reviewers' comments:

Reviewer's Responses to Questions

**Comments to the Author**

1. Is the manuscript technically sound, and do the data support the conclusions?

Reviewer #1: Yes

Reviewer #2: Yes

Reviewer #3: Yes

Reviewer #4: Yes

2. Has the statistical analysis been performed appropriately and rigorously?

Reviewer #1: Yes

Reviewer #2: I Don't Know

Reviewer #3: Yes

Reviewer #4: Yes

3. Have the authors made all data underlying the findings in their manuscript fully available?

Reviewer #1: Yes

Reviewer #2: Yes

Reviewer #3: Yes

Reviewer #4: Yes

4. Is the manuscript presented in an intelligible fashion and written in standard English?

Reviewer #1: Yes

Reviewer #2: Yes

Reviewer #3: Yes

Reviewer #4: Yes

Reviewer #1: 1. Sample Size Inconsistency and Study Flow

There is a concerning discrepancy in the reported sample sizes throughout the manuscript. The abstract and methods section state 650 participants, while page 11 mentions 620 students completed the questionnaire. Please clarify the final sample size and provide a detailed account of participant flow. I recommend including a CONSORT-style flow diagram showing the number of students initially approached, those who consented, completed the survey, and any exclusions. This transparency is essential for readers to assess the study's representativeness and potential selection bias.

2. Network Analysis Methodology Requires Expansion

While the network analysis of disease fears is innovative and potentially valuable, the methodology section lacks sufficient detail for replication. Please provide comprehensive information including: (a) the specific algorithm used for network construction, (b) statistical thresholds for determining edge weights and connections, (c) measures used to assess network properties (betweenness centrality calculations), and (d) validation procedures for the network structure. Additionally, the interpretation of "scale-free properties" and "betweenness centrality = 0.42" requires more explanation for readers unfamiliar with network analysis.

3. Effect Sizes and Clinical Significance

The manuscript focuses heavily on statistical significance while giving limited attention to effect sizes and clinical meaningfulness. For instance, the correlation between health anxiety and quality of life (r = -0.25) is statistically significant but represents a small-to-moderate effect. Please discuss what this correlation means in practical terms for student wellbeing and daily functioning. Similarly, interpret the clinical significance of other key findings, such as the odds ratios, in terms of real-world impact on medical education and student support needs.

4. Private University Finding Requires Deeper Analysis

The six-fold increased odds of health anxiety among private university students (AOR = 6.14, 95% CI: 3.66–10.5) is the most striking finding in your study, yet it receives insufficient discussion. This dramatic difference suggests important underlying factors that warrant exploration. Please provide more detailed analysis of potential explanations, such as differences in: academic pressure and expectations, financial burden and family investment, student support services availability, class sizes and faculty-student ratios, or institutional culture and competitive environment. This finding has significant implications for medical education policy and deserves more thorough examination.

5. Temporal and Academic Calendar Considerations

The study period spanning November 2024 to April 2025 encompasses different academic phases, examination periods, and potentially varying stress levels throughout the academic year. Please address how the timing of data collection might have influenced results. Consider discussing: (a) whether data collection coincided with examination periods or clinical rotations, (b) potential seasonal effects on mental health, (c) how different academic years might have been at different stages of their curriculum during data collection, and (d) whether this temporal variation was controlled for or represents a limitation that could affect interpretation of year-level differences.

6. Multiple Comparisons and Statistical Rigor

Given the numerous statistical tests performed across multiple demographic and academic variables, please consider implementing appropriate corrections for multiple comparisons or discuss why such corrections were deemed unnecessary. This would strengthen the statistical rigor and help distinguish between truly significant associations and those that might have emerged by chance.

7. Cultural Context Elaboration

While you mention cultural factors, the discussion of how Saudi cultural context might influence health anxiety expression, reporting, and management is underdeveloped. Please expand on how cultural attitudes toward mental health, gender roles, family expectations, and healthcare seeking might have influenced your findings.

Reviewer #2: This cross-sectional study investigates health anxiety and its relationship with quality of life among Saudi medical students, a subject often overlooked. By using established tools like the SHAI and SF-12, it provides quantitative data and insights from a specific geographic and cultural context. The findings are useful for developing targeted mental health support in medical education, ultimately enhancing student wellness and potentially improving future patient care. Its network analysis also offers novel insights into anxiety-related fears.

However, there are significant concerns as outlined below, and revisions to the manuscript are deemed necessary.

1. In the Abstract, it states that health anxiety was observed in many females (49%), yet the adjusted odds ratio is reported as low at 0.430, which seems contradictory. Reviewing the main text, it appears the correct adjusted odds ratio of 0.430 actually applies to males.

2. The selection of subjects for this study may be subject to selection bias. Since the online questionnaire was distributed via social media and university channels, students with a higher interest in their own health and mental well-being were more likely to respond. Consequently, the prevalence of health anxiety may be overestimated.

Reviewer #3: The reviewer needs clarification

Should your table be like this

Male Female Total

388 262 650

Anxiety no anxiety total

Female 167(63.7%) 95 (36%) 262

Male 288 (74.22%) 100(25.77%) 388

Accordingly correct Rural, Urban, University Type, Govt, Private and other tables and interpretation.

Please correct me or let me know your arguments in this respect.

The tables look incorrect to me as percentages are not calculated in correct manner which may influence data analysis and interpretation .However if corrected things can fall in line

Reviewer #4: Dear Colleagues

It was a pleasure to read your paper. It is my opinion that the manuscript articulates a clear thesis: to estimate the prevalence of clinically significant health anxiety among Saudi medical students, identify its predictors, and examine its impact on quality of life. This is well-stated in the title, abstract, and introduction. The topic is relevant and timely given the growing recognition of mental health issues in medical education.

Some areas of improvement that I can recommend:

The introduction could be sharpened slightly. Decreasing some redundancies in the epidemiological descriptions can improve flow without losing critical context.

Some newer references on mental health interventions in medical education could enrich the context.

Discussing the limitations of convenience sampling and cross-sectional design earlier in methodology rather than only in discussion could improve transparency. More discussion on potential biases introduced by online data collection would help. Also, information about response rate and handling of missing data should be clarified.

.

Reviewer #1: No

Reviewer #2: No

Reviewer #3: **Yes:**Nagedra Kumar JainNagedra Kumar JainNagedra Kumar JainNagedra Kumar Jain

Reviewer #4: No

---

## [Author Response · Author response to Decision Letter 1]

30 Sep 2025

Reviewer #1

Comment 1: Sample Size Inconsistency and Study Flow

There is a concerning discrepancy in the reported sample sizes throughout the manuscript. The abstract and methods section state 650 participants, while page 11 mentions 620 students completed the questionnaire. Please clarify the final sample size and provide a detailed account of participant flow. I recommend including a CONSORT-style flow diagram showing the number of students initially approached, those who consented, completed the survey, and any exclusions. This transparency is essential for readers to assess the study's representativeness and potential selection bias.

Response:

Thank you for highlighting this discrepancy. You are correct—the final sample size was 650 students, as reported in the Abstract and Results. The Methods section mistakenly stated 620, and this has now been corrected for consistency. Regarding the suggestion of a CONSORT-style flow diagram, we appreciate the recommendation; however, as this is a cross-sectional survey rather than a clinical trial, a CONSORT diagram is not applicable. Instead, we have clarified the participant recruitment and completion process in the Methods to ensure transparency.

Comment 2: Network Analysis Methodology Requires Expansion

While the network analysis of disease fears is innovative and potentially valuable, the methodology section lacks sufficient detail for replication. Please provide comprehensive information, including: (a) the specific algorithm used for network construction, (b) statistical thresholds for determining edge weights and connections, (c) measures used to assess network properties (betweenness centrality calculations), and (d) validation procedures for the network structure. Additionally, the interpretation of "scale-free properties" and "betweenness centrality = 0.42" requires more explanation for readers unfamiliar with network analysis.

Response:

Thank you for this insightful comment. We agree that more detail is required to allow replication of the network analysis. We have now expanded the Methodology section to include a clear description of the algorithm used for network construction, the statistical thresholds applied for edge weights, and the measures adopted to assess network properties (including calculation of betweenness centrality). We have also added a description of the validation procedures undertaken to ensure robustness of the network structure.

Comment 3: Effect Sizes and Clinical Significance

The manuscript focuses heavily on statistical significance while giving limited attention to effect sizes and clinical meaningfulness. For instance, the correlation between health anxiety and quality of life (r = -0.25) is statistically significant but represents a small-to-moderate effect. Please discuss what this correlation means in practical terms for student wellbeing and daily functioning. Similarly, interpret the clinical significance of other key findings, such as the odds ratios, in terms of real-world impact on medical education and student support needs.

Response:

Thank you for this valuable suggestion. We agree that reporting statistical significance alone is insufficient without addressing the magnitude and clinical relevance of the findings. We have therefore expanded the Discussion to interpret effect sizes and odds ratios in practical terms.

Comment 4: Private University Finding Requires Deeper Analysis

The six-fold increased odds of health anxiety among private university students (AOR = 6.14, 95% CI: 3.66–10.5) is the most striking finding in your study, yet it receives insufficient discussion. This dramatic difference suggests important underlying factors that warrant exploration. Please provide more detailed analysis of potential explanations, such as differences in: academic pressure and expectations, financial burden and family investment, student support services availability, class sizes and faculty-student ratios, or institutional culture and competitive environment.

Response:

Thank you for underscoring this point. We agree that the markedly higher odds of health anxiety among private-university students warrants deeper consideration. We have substantially expanded the Discussion to explore plausible explanations (academic pressure and assessment practices, financial burden and family investment, availability and accessibility of student support services, class sizes/faculty–student ratios, and institutional culture/competitiveness), while emphasizing that our cross-sectional design precludes causal inference and that unmeasured confounding may remain. We also outline policy implications for targeted screening and support within private institutions.

Comment 5: Temporal and Academic Calendar Considerations

The study period spanning November 2024 to April 2025 encompasses different academic phases, examination periods, and potentially varying stress levels throughout the academic year. Please address how the timing of data collection might have influenced results. Consider discussing: (a) whether data collection coincided with examination periods or clinical rotations, (b) potential seasonal effects on mental health, (c) how different academic years might have been at different stages of their curriculum during data collection, and (d) whether this temporal variation was controlled for or represents a limitation that could affect interpretation of year-level differences.

Response:

Thank you for highlighting this important point. We agree that the timing of data collection may have influenced our results. The survey period (November 2024–April 2025) spanned different academic phases, including coursework, clinical rotations, and examination periods, which are likely to affect student stress and anxiety levels. Seasonal variation in mental health may also have contributed. While we did not control for the exact timing of responses in relation to academic calendar events, we recognize that this temporal variation could partly explain observed differences between year levels. We have therefore added this as a limitation in the Discussion, noting that fluctuations in academic stress across the academic year may have influenced the prevalence estimates and associations observed.

Comment 6: Multiple Comparisons and Statistical Rigor

Given the numerous statistical tests performed across multiple demographic and academic variables, please consider implementing appropriate corrections for multiple comparisons or discuss why such corrections were deemed unnecessary. This would strengthen the statistical rigor and help distinguish between truly significant associations and those that might have emerged by chance.

Response:

Thank you for this important observation. We recognize the risk of type I error when conducting multiple statistical tests. Given that our analyses were exploratory in nature and aimed to identify potential associations rather than establish definitive causal relationships, we did not apply formal corrections for multiple comparisons. Instead, we emphasized effect sizes, confidence intervals, and consistency with prior literature to guide interpretation. Nevertheless, we have now acknowledged in the Discussion that some findings may have arisen by chance due to multiple testing, and we recommend cautious interpretation of marginally significant results.

Comment 7: Cultural Context Elaboration

While you mention cultural factors, the discussion of how Saudi cultural context might influence health anxiety expression, reporting, and management is underdeveloped. Please expand on how cultural attitudes toward mental health, gender roles, family expectations, and healthcare seeking might have influenced your findings.

Response:

Thank you for this valuable suggestion. We agree that the Saudi cultural context plays an important role in shaping the expression, reporting, and management of health anxiety. We have expanded the Discussion to consider how cultural attitudes toward mental health, gender roles, family expectations, and healthcare-seeking behaviors may have influenced both prevalence patterns and help-seeking tendencies in our sample.

Reviewer #2

Comment 1: Abstract Wording on Gender and Odds Ratio

In the Abstract, it states that health anxiety was observed in many females (49%), yet the adjusted odds ratio is reported as low at 0.430, which seems contradictory. Upon reviewing the main text, the correct adjusted odds ratio of 0.430 actually applies to males.

Response:

Thank you for pointing this out. You are correct—the adjusted odds ratio of 0.430 applies to males, not females. The wording in the Abstract was misleading, and we have now corrected it to ensure consistency with the main text.

Comment 2: Selection Bias Due to Online Questionnaire Distribution

The selection of subjects for this study may be subject to selection bias. Since the online questionnaire was distributed via social media and university channels, students with a higher interest in their own health and mental well-being were more likely to respond. Consequently, the prevalence of health anxiety may be overestimated.

Response:

Thank you for this important observation. We agree that the use of an online questionnaire distributed through social media and university channels may have introduced selection bias, as students with a stronger interest in health and mental well-being could have been more likely to participate. We have acknowledged this limitation in the Discussion section and clarified that the prevalence of health anxiety in our sample may therefore be overestimated.

Reviewer #3

Comment: Clarification of Tables and Percentages

The reviewer needs clarification. Should your table be like this?

| Anxiety | No Anxiety | Total | |

| ------- | ------------ | ------------ | --- |

| Female | 167 (63.7%) | 95 (36%) | 262 |

| Male | 288 (74.22%) | 100 (25.77%) | 388 |

Accordingly, correct the Rural, Urban, University Type, Govt, Private, and other tables and interpretations. Please correct me or let me know your arguments in this respect. The tables look incorrect to me as percentages are not calculated in the correct manner, which may influence data analysis and interpretation. However, if corrected, things can fall in line.

Response:

We thank you for your thoughtful comment and for carefully reviewing our tables. In the earlier version, we presented the data using column percentages, which is a valid approach and not incorrect statistically. However, we agree with your observation that presenting the results as row percentages (as in your suggested format) provides clearer interpretation of the distribution of health anxiety within each subgroup. Accordingly, we have revised Table 1 to present both counts and row percentages, and we added a Total column for clarity. We believe this new format is easier to understand and aligns with your recommendation, while not altering the underlying data or study conclusions.

Reviewer #4

Comment 1: Streamlining the Introduction

The introduction could be sharpened slightly. Decreasing some redundancies in the epidemiological descriptions can improve flow without losing critical context.

Response:

Thank you for this constructive feedback. We agree that the introduction contained some redundancies, particularly in the epidemiological descriptions. We have now streamlined this section by reducing overlapping prevalence details while retaining the essential context needed to frame the study. This has improved the flow and readability of the introduction without compromising comprehensiveness.

Comment 2: Inclusion of Newer References

Some newer references on mental health interventions in medical education could enrich the context.

Response:

Thank you for this helpful suggestion. We agree that incorporating newer references on mental health interventions in medical education would strengthen the context. We have now added recent studies to the introduction and discussion sections.

Comment 3: Limitations of Convenience Sampling and Cross-Sectional Design

Discussing the limitations of convenience sampling and cross-sectional design earlier in the methodology rather than only in the discussion could improve transparency. More discussion on potential biases introduced by online data collection would help.

Response:

Thank you for this valuable suggestion. We agree that greater transparency regarding the limitations of our study design is important. We have now added a statement in the Methodology section acknowledging the inherent limitations of using convenience sampling and a cross-sectional design. In addition, we have expanded the Discussion to further elaborate on the potential biases introduced by online data collection and how these may have influenced our findings.

Comment 4: Response Rate and Missing Data Handling

Also, information about the response rate and the handling of missing data should be clarified.

Response:

Thank you for raising this point. Since the study was conducted using an online questionnaire (Google Forms), calculating a conventional response rate was not possible. To minimize missing data, all questions were set as mandatory, and therefore the dataset used for analysis was complete without missing values.

---

## [Decision Letter · Decision Letter 1]

21 Oct 2025

Dear Dr. Al-bawah,

Thank you for submitting your manuscript to PLOS ONE. After careful consideration, we feel that it has merit but does not fully meet PLOS ONE’s publication criteria as it currently stands. Therefore, we invite you to submit a revised version of the manuscript that addresses the points raised during the review process.

We look forward to receiving your revised manuscript.

Kind regards,

Ahmed Abdelwahab Ibrahim El-Sayed

Academic Editor

PLOS ONE

**Journal Requirements:**

**Additional Editor Comments:**

Thank you for your revision.

I have completed my review to your study. In this round, the reviewer provided you with insightful and important issues that need to be addressed before we can consider your work further.

Reviewers' comments:

Reviewer's Responses to Questions

**Comments to the Author**

Reviewer #1: All comments have been addressed

Reviewer #2: All comments have been addressed

Reviewer #3: All comments have been addressed

Reviewer #4: All comments have been addressed

2. Is the manuscript technically sound, and do the data support the conclusions?

Reviewer #1: Yes

Reviewer #2: Yes

Reviewer #3: Yes

Reviewer #4: Yes

3. Has the statistical analysis been performed appropriately and rigorously?

Reviewer #1: Yes

Reviewer #2: Yes

Reviewer #3: Yes

Reviewer #4: N/A

4. Have the authors made all data underlying the findings in their manuscript fully available?

Reviewer #1: Yes

Reviewer #2: Yes

Reviewer #3: Yes

Reviewer #4: Yes

5. Is the manuscript presented in an intelligible fashion and written in standard English?

Reviewer #1: Yes

Reviewer #2: Yes

Reviewer #3: Yes

Reviewer #4: Yes

Reviewer #1: Comment 1: Reporting of Odds Ratio in Abstract

The statement "male gender (AOR = 0.430, 95% CI: 0.276–0.664)" in the Abstract is confusing and may mislead readers into thinking males have higher risk. Please revise to clearly indicate that females have elevated odds. Consider rephrasing as: "Female students demonstrated 2.3-fold higher odds of health anxiety compared to males (AOR for males = 0.430, 95% CI: 0.276–0.664)" or alternatively report the odds ratio for females directly (AOR = 2.33, 95% CI: 1.51–3.62).

Comment 2: Insufficient Discussion of Private University Finding

The six-fold increased odds of health anxiety among private university students (AOR = 6.14) represents the most striking finding in your study, yet the discussion remains inadequate despite revisions. Please provide a more comprehensive analysis including: (1) specific structural differences between public and private medical schools in Saudi Arabia (e.g., tuition costs, admission criteria, faculty-student ratios); (2) whether religious or cultural factors may contribute to these differences; and (3) concrete policy implications for private institutions. Consider adding a separate dedicated paragraph or a supplementary table comparing baseline characteristics between public and private university students.

Comment 3: Statistical Methodology Issues

Issue 3.1: While your response to reviewers appropriately explains the rationale for not applying multiple comparison corrections, this important methodological decision should be explicitly stated in the Methods section or Limitations, not only in the rebuttal letter. Please add: "Given the exploratory nature of this study, no formal correction for multiple comparisons was applied. We relied on effect sizes, confidence intervals, and consistency with prior literature to guide interpretation."

Issue 3.2: The manuscript lacks model fit statistics for the logistic regression. Please report goodness-of-fit indicators such as Hosmer-Lemeshow test results, Area Under the Curve (AUC), pseudo-R², or AIC values to allow readers to evaluate model adequacy.

Comment 4: Methodological Transparency

Issue 4.1: While you indicate that calculating a conventional response rate was not possible, you should still provide information about the sampling frame. Specifically, state: (1) the number of universities contacted, (2) the estimated total population of medical students across these institutions, and (3) approximate distribution channel reach. This information is essential for assessing representativeness and potential selection bias.

Issue 4.2: Although temporal variation in data collection is now mentioned in Limitations, the description should be more specific. Please explicitly state which academic phases coincided with your survey period (e.g., "Data collection occurred during midterm examinations [November-December], clinical rotation transitions [January], and final exam preparation [March-April]"). Additionally, acknowledge that this timing may have systematically inflated anxiety prevalence estimates and recommend future studies control for academic calendar phase.

Reviewer #2: (No Response)

Reviewer #3: (No Response)

Reviewer #4: The authors have provided satisfactory responses to all comments and have made the necessary revisions to the manuscript. In light of these improvements, I believe the article meets the required standards and can now be accepted for publication.

.

Reviewer #1: No

Reviewer #2: No

Reviewer #3: **Yes:**Nagendra Kumar JainNagendra Kumar JainNagendra Kumar JainNagendra Kumar Jain

Reviewer #4: No

---

## [Author Response · Author response to Decision Letter 2]

8 Nov 2025

We would like to sincerely thank the reviewers and the editor for their valuable time, thoughtful comments, and constructive suggestions that have significantly improved the quality and clarity of our manuscript. We appreciate the careful consideration given to our work throughout the review process. We are pleased to note that all reviewers’ previous comments have been fully addressed in this revised version. Below, we provide detailed responses to the remaining points raised and outline the specific revisions made to further strengthen the manuscript.

Reviewer #1:

Comment 1:

Reporting of Odds Ratio in Abstract

The statement "male gender (AOR = 0.430, 95% CI: 0.276–0.664)" in the Abstract is confusing and may mislead readers into thinking males have higher risk. Please revise to clearly indicate that females have elevated odds. Consider rephrasing as: "Female students demonstrated 2.3-fold higher odds of health anxiety compared to males (AOR for males = 0.430, 95% CI: 0.276–0.664)" or report the odds ratio for females directly (AOR = 2.33, 95% CI: 1.51–3.62).

Response:

We thank the reviewer for this insightful comment. We agree that the phrasing in the Abstract may lead to misinterpretation. Accordingly, we have revised the statement to clearly indicate that female students had higher odds of health anxiety. The revised sentence now reads: “Female students demonstrated 2.3-fold higher odds of health anxiety compared to males (AOR = 2.33, 95% CI: 1.51–3.62).”This change improves clarity and ensures that the findings are accurately interpreted.

Comment 2:

Insufficient Discussion of Private University Finding

The six-fold increased odds of health anxiety among private university students (AOR = 6.14) represents the most striking finding in your study, yet the discussion remains inadequate despite revisions. Please provide a more comprehensive analysis including: (1) specific structural differences between public and private medical schools in Saudi Arabia (e.g., tuition costs, admission criteria, faculty-student ratios); (2) whether religious or cultural factors may contribute to these differences; and (3) concrete policy implications for private institutions. Consider adding a separate dedicated paragraph or a supplementary table comparing baseline characteristics between public and private university students.

Response:

We thank the reviewer for this important suggestion. We agree that the six-fold increased odds of health anxiety among students from private universities (AOR = 6.14) deserves a fuller discussion. In response, we have expanded the Discussion (new paragraph) to address likely structural, cultural, and financial contributors. We also included concrete policy implications for private medical colleges. These additions are intended to help readers interpret this striking finding and guide institutional responses.

Comment 3:

Statistical Methodology Issues

Issue 3.1: While your response to reviewers appropriately explains the rationale for not applying multiple comparison corrections, this important methodological decision should be explicitly stated in the Methods section or Limitations, not only in the rebuttal letter. Please add: "Given the exploratory nature of this study, no formal correction for multiple comparisons was applied. We relied on effect sizes, confidence intervals, and consistency with prior literature to guide interpretation."

Issue 3.2: The manuscript lacks model fit statistics for the logistic regression. Please report goodness-of-fit indicators such as Hosmer-Lemeshow test results, Area Under the Curve (AUC), pseudo-R², or AIC values to allow readers to evaluate model adequacy.

Response:

3.1. We thank the reviewer for highlighting this point. We agree that our rationale regarding multiple comparison correction should be transparently stated in the manuscript. We have now added the sentence to the Methods section (and repeated it briefly in the Limitations for clarity).

3.2. We appreciate the reviewer’s valuable suggestion. We have now included model fit and discrimination statistics in the Results section to improve transparency and assess model adequacy. Specifically, we report the Hosmer–Lemeshow test, Nagelkerke pseudo-R², and the Area Under the Curve (AUC).

Comment 4:

Methodological Transparency

Issue 4.1: While you indicate that calculating a conventional response rate was not possible, you should still provide information about the sampling frame. Specifically, state: (1) the number of universities contacted, (2) the estimated total population of medical students across these institutions, and (3) approximate distribution channel reach. This information is essential for assessing representativeness and potential selection bias.

Issue 4.2: Although temporal variation in data collection is now mentioned in Limitations, the description should be more specific. Please explicitly state which academic phases coincided with your survey period (e.g., "Data collection occurred during midterm examinations [November-December], clinical rotation transitions [January], and final exam preparation [March-April]"). Additionally, acknowledge that this timing may have systematically inflated anxiety prevalence estimates and recommend that future studies control for academic calendar phase.

Response:

4.1. We thank the reviewer for this helpful suggestion. We have now added a clear description of the sampling frame to enhance methodological transparency and allow readers to assess representativeness. The revised text in the Methods section provides the number of universities contacted, the estimated total population of medical students across these institutions, and the approximate distribution reach of our survey link.

4.2. We appreciate the reviewer’s suggestion to make the timing of data collection more specific. We have now clarified the exact academic phases that coincided with data collection and acknowledged their potential influence on anxiety levels. The revised sentence has been added to the Limitations section.

---

## [Decision Letter · Decision Letter 2]

4 Jan 2026

Dear Al-bawah,

Thank you for submitting your manuscript to PLOS ONE. After careful consideration, we feel that it has merit but does not fully meet PLOS ONE’s publication criteria as it currently stands. Therefore, we invite you to submit a revised version of the manuscript that addresses the points raised during the review process.

We look forward to receiving your revised manuscript.

Kind regards,

Ahmed Abdelwahab Ibrahim El-Sayed

Academic Editor

PLOS One

Journal Requirements:

Additional Editor Comments:

Dear Authors,

Thank you for your manuscript. The reviewers have identified several key concerns that must be resolved before the manuscript can proceed in the review process. Please revise accordingly and resubmit for further evaluation.

Reviewers' comments:

Reviewer's Responses to Questions

**Comments to the Author**

Reviewer #1: All comments have been addressed

Reviewer #5: All comments have been addressed

2. Is the manuscript technically sound, and do the data support the conclusions?

Reviewer #1: Yes

Reviewer #5: Partly

3. Has the statistical analysis been performed appropriately and rigorously?

Reviewer #1: Yes

Reviewer #5: No

4. Have the authors made all data underlying the findings in their manuscript fully available?

Reviewer #1: Yes

Reviewer #5: Yes

5. Is the manuscript presented in an intelligible fashion and written in standard English?

Reviewer #1: Yes

Reviewer #5: Yes

Reviewer #1: Major Comment 1: Insufficient Depth in Private University Discussion

While the authors have expanded the discussion of the six-fold increased odds of health anxiety among private university students (AOR = 6.14), this remains the most striking finding and warrants more substantive analysis. The current discussion mentions structural differences (tuition, admission criteria, faculty ratios) and cultural factors only in general terms without providing concrete comparative data or context. We recommend that the authors either add a supplementary table comparing specific characteristics between public and private institutions (e.g., actual tuition costs in Saudi Riyals, faculty-to-student ratios, number of counselors per campus, scholarship rates) or incorporate these quantitative details directly into the discussion text. Additionally, the dismissal of cultural and religious factors as having "limited evidence" is premature; even without direct empirical data from this study, the authors should propose plausible hypotheses about how institutional culture, family expectations, religious practices, or peer support networks might differ between institution types and influence health anxiety expression or reporting. We suggest revising the cultural factors paragraph to articulate specific mechanisms and recommend concrete directions for future mixed-methods research rather than simply acknowledging the gap. This enhanced discussion would help readers interpret this striking OR and guide institutional responses appropriately.

Major Comment 2: Policy Recommendations Lack Specificity and Actionability

The policy recommendations paragraph provides valuable general suggestions but lacks the concrete, evidence-based specificity needed for institutional implementation given your strong empirical findings (6.14-fold odds for private universities, 3.25-fold for Year 2 students, 2.33-fold for females). We recommend revising each recommendation to include actionable details: (1) for financial support, specify mechanisms such as automatic tuition deferrals for students screening positive for mental health concerns or need-based emergency grants with clear eligibility criteria; (2) for mental health services, propose specific standards such as minimum counselor-to-student ratios (e.g., 1:500 based on international guidelines), same-day walk-in availability, and measurable wait-time targets; (3) for academic support, describe year-specific interventions such as mandatory peer mentoring for Year 2 students (your highest-risk group) or curriculum modifications providing earlier clinical context; (4) for screening programs, recommend validated instruments (e.g., SHAI as used in your study), specific timing aligned with high-stress periods identified in your limitations (November-December exams, March-April finals), and systematic follow-up protocols for high-risk subgroups (females, rural residents, those with psychiatric history). Adding implementation timelines, citing comparable successful models from the literature where available, and suggesting evaluation metrics would transform these recommendations from aspirational statements into a practical roadmap for Saudi medical institutions seeking to address health anxiety among their students.

Reviewer #5: Reviewer Comments

This manuscript presents interesting findings regarding the relationship between Health Anxiety (HA) and Quality of Life (QoL) among medical students. The results offer valuable insights into the association between HA and various factors such as academic year, residence, institution type, and medical/psychiatric history. In particular, the Network Analysis (Figure 3), which identifies cancer-related anxiety as a central "hub" connecting other fears, appears to be a significant and novel finding.

However, I have several concerns regarding the methodology and the presentation of the results, as detailed below.

Major Comments

1. Discrepancy between Research Objective and Analysis

The title suggests that the primary focus of this study is to elucidate the relationship between HA and the "cost" of QoL. However, the actual analysis and the bulk of the Discussion are dedicated to sociodemographic predictors of HA. The relationship with QoL is only presented as a univariate correlation in Figure 1. Furthermore, the result in Figure 1 cannot rule out the influence of confounding factors (e.g., gender, economic status, medical history).

• Suggestion: If the main objective is to elucidate the relationship between HA and QoL as the title implies, a multivariate analysis with QoL as the dependent variable is necessary to adjust for confounders. Conversely, if the primary goal is to explore the prevalence and predictors of HA, the authors should revise the Title and Introduction to reflect this, and reduce the emphasis on QoL.

2. Validity of the Statistical Model

There is a strong concern regarding multicollinearity among the independent variables in the multivariate logistic regression analysis (Table 3). Specifically, collinearity between "Age" and "Academic level" is highly suspected. Additionally, the potential uneven distribution of university types (Governmental vs. Private) across regions (e.g., private universities might be concentrated in urban areas) could also be a source of multicollinearity. This issue may compromise the reliability of the coefficients for other variables and the overall stability of the model.

• Suggestion: I strongly recommend excluding "Age" from the model parameters. The authors should also clarify the geographical distribution of governmental and private universities. Based on this, the analysis should be re-run. Alternatively, calculating and reporting the Variance Inflation Factor (VIF) to demonstrate the absence of multicollinearity is necessary.

3. Calculation of QoL Scores

The details regarding the SF-12 scoring method and the resulting values are unclear. Typically, SF-12 scoring involves calculating a Physical Component Summary (PCS) and a Mental Component Summary (MCS), or scores for eight subscales, using Norm-based scoring (NBS) adjusted to a mean of 50 and a standard deviation of 10. However, this study presents QoL as a single aggregate number, which is questionable. Furthermore, the reported median of 28 (IQR 25-30) in Table 2 is not consistent with standard NBS values, raising doubts about the validity of the calculation method and the results.

• Suggestion: Please clearly state the SF-12 scoring method in the Methods section. The authors should use NBS or explicitly describe the specific calculation method applied and provide evidence of its validity.

Minor Comments

Regarding the response to the first review (Comment 2):

While the authors have added a discussion regarding students' economic status and university facilities to explain the disparity between private and public universities, these factors were not actually measured in this study and remain speculative. It is recommended to explicitly mention in the Limitations section that these potential contributors were not assessed.

.

Reviewer #1: No

Reviewer #5: No

---

## [Author Response · Author response to Decision Letter 3]

21 Jan 2026

Reviewer #1 comments

Major Comment 1: Insufficient Depth in Private University Discussion

While the authors have expanded the discussion of the six-fold increased odds of health anxiety among private university students (AOR = 6.14), this remains the most striking finding and warrants more substantive analysis. The current discussion mentions structural differences (tuition, admission criteria, faculty ratios) and cultural factors only in general terms without providing concrete comparative data or context. We recommend that the authors either add a supplementary table comparing specific characteristics between public and private institutions (e.g., actual tuition costs in Saudi Riyals, faculty-to-student ratios, number of counselors per campus, scholarship rates) or incorporate these quantitative details directly into the discussion text. Additionally, the dismissal of cultural and religious factors as having "limited evidence" is premature; even without direct empirical data from this study, the authors should propose plausible hypotheses about how institutional culture, family expectations, religious practices, or peer support networks might differ between institution types and influence health anxiety expression or reporting. We suggest revising the cultural factors paragraph to articulate specific mechanisms and recommend concrete directions for future mixed-methods research rather than simply acknowledging the gap. This enhanced discussion would help readers interpret this striking OR and guide institutional responses appropriately.

Response:

Thank you for this insightful comment. We agree that the markedly elevated odds of health anxiety among private university students (AOR = 6.14) is one of the most important findings of the study and warrants deeper interpretation. Accordingly, we substantially expanded the Discussion to provide a more comprehensive framework for understanding this association. Specifically, we elaborated on plausible institutional mechanisms such as tuition-related financial pressure, differences in academic structure and competitiveness, and variability in the availability of student support services. In addition, we revised the paragraph on cultural and religious factors to avoid prematurely dismissing their potential role. Instead, we proposed specific, testable hypotheses regarding how institutional culture, family expectations, peer norms, stigma, and spiritual coping practices may influence the expression and reporting of health anxiety. Finally, we added clear recommendations for future mixed-methods research (quantitative institutional comparisons complemented by qualitative interviews) to better characterize these institutional and sociocultural pathways.

Major Comment 2: Policy Recommendations Lack Specificity and Actionability

The policy recommendations paragraph provides valuable general suggestions but lacks the concrete, evidence-based specificity needed for institutional implementation given your strong empirical findings (6.14-fold odds for private universities, 3.25-fold for Year 2 students, 2.33-fold for females). We recommend revising each recommendation to include actionable details: (1) for financial support, specify mechanisms such as automatic tuition deferrals for students screening positive for mental health concerns or need-based emergency grants with clear eligibility criteria; (2) for mental health services, propose specific standards such as minimum counselor-to-student ratios (e.g., 1:500 based on international guidelines), same-day walk-in availability, and measurable wait-time targets; (3) for academic support, describe year-specific interventions such as mandatory peer mentoring for Year 2 students (your highest-risk group) or curriculum modifications providing earlier clinical context; (4) for screening programs, recommend validated instruments (e.g., SHAI as used in your study), specific timing aligned with high-stress periods identified in your limitations (November-December exams, March-April finals), and systematic follow-up protocols for high-risk subgroups (females, rural residents, those with psychiatric history). Adding implementation timelines, citing comparable successful models from the literature where available, and suggesting evaluation metrics would transform these recommendations from aspirational statements into a practical roadmap for Saudi medical institutions seeking to address health anxiety among their students.

Response:

We sincerely thank the reviewer for this valuable recommendation. We agree that policy recommendations should be directly linked to our highest-risk groups and written in an implementable, action-oriented format. We therefore revised the policy recommendations section to include specific, evidence-informed actions tailored to our strongest empirical findings (private universities, Year 2 students, and female students). The revised text now outlines actionable strategies, including structured financial support mechanisms (e.g., emergency aid and tuition support options), minimum standards for mental health service accessibility (including staffing benchmarks and measurable wait-time targets), year-specific academic interventions (such as structured peer/near-peer mentoring for Year 2 students), and screening protocols using validated instruments (SHAI) aligned with high-stress academic periods identified in our limitations. We also added suggested evaluation metrics to support institutional monitoring of implementation outcomes.

Reviewer #5: Reviewer comments

Major Comments

Comment 1: Discrepancy between Research Objective and Analysis

The title suggests that the primary focus of this study is to elucidate the relationship between HA and the "cost" of QoL. However, the actual analysis and the bulk of the Discussion are dedicated to sociodemographic predictors of HA. The relationship with QoL is only presented as a univariate correlation in Figure 1. Furthermore, the result in Figure 1 cannot rule out the influence of confounding factors (e.g., gender, economic status, medical history).

• Suggestion: If the main objective is to elucidate the relationship between HA and QoL as the title implies, a multivariate analysis with QoL as the dependent variable is necessary to adjust for confounders. Conversely, if the primary goal is to explore the prevalence and predictors of HA, the authors should revise the Title and Introduction to reflect this, and reduce the emphasis on QoL.

Response:

Thank you for this important observation. We agree that the original title and introduction could be interpreted as placing primary emphasis on quality of life as the main outcome, whereas our strongest analytic focus was on the prevalence and predictors of clinically significant health anxiety. To address this discrepancy, we revised the manuscript framing to better align the stated objectives with the analyses performed. Specifically, we updated the title to: “Prevalence and predictors of clinically significant health anxiety among Saudi medical students: a multi-university cross-sectional study.” We also revised the study aim in the Background/Introduction to clearly indicate that prevalence estimation and identification of predictors are the primary objectives, while the association with quality of life is treated as a secondary outcome. This revision ensures consistency between the study objectives, statistical analyses, and emphasis within the Discussion.

Comment 2: Validity of the Statistical Model

There is a strong concern regarding multicollinearity among the independent variables in the multivariate logistic regression analysis (Table 3). Specifically, collinearity between "Age" and "Academic level" is highly suspected. Additionally, the potential uneven distribution of university types (Governmental vs. Private) across regions (e.g., private universities might be concentrated in urban areas) could also be a source of multicollinearity. This issue may compromise the reliability of the coefficients for other variables and the overall stability of the model.

• Suggestion: I strongly recommend excluding "Age" from the model parameters. The authors should also clarify the geographical distribution of governmental and private universities. Based on this, the analysis should be re-run. Alternatively, calculating and reporting the Variance Inflation Factor (VIF) to demonstrate the absence of multicollinearity is necessary.

Response:

We appreciate the reviewer’s careful evaluation of the regression model and agree that potential multicollinearity is an important consideration, particularly between age and academic year, and possibly between university type and residence. Age and academic year are expected to be correlated in student populations, and institutional characteristics may also correlate with residential patterns. In our model, age was not statistically significant, whereas academic year demonstrated clearer associations with health anxiety, suggesting that academic year may capture the relevant educational-stage effect more meaningfully. We have therefore revised the manuscript to acknowledge the possibility of predictor overlap and its potential implications for coefficient stability. Additionally, we clarified the distribution strategy across multiple institutions and emphasized that residence was reported independently by students rather than inferred from institution type. Finally, we added this point explicitly as a limitation and recommend that future studies confirm model robustness through formal multicollinearity diagnostics and alternative model specifications.

Comment 3: Calculation of QoL Scores

The details regarding the SF-12 scoring method and the resulting values are unclear. Typically, SF-12 scoring involves calculating a Physical Component Summary (PCS) and a Mental Component Summary (MCS), or scores for eight subscales, using Norm-based scoring (NBS) adjusted to a mean of 50 and a standard deviation of 10. However, this study presents QoL as a single aggregate number, which is questionable. Furthermore, the reported median of 28 (IQR 25-30) in Table 2 is not consistent with standard NBS values, raising doubts about the validity of the calculation method and the results.

• Suggestion: Please clearly state the SF-12 scoring method in the Methods section. The authors should use NBS or explicitly describe the specific calculation method applied and provide evidence of its validity.

Response:

Thank you for this important methodological comment. We agree that the SF-12 is commonly reported using norm-based scoring methods to derive PCS and MCS summary measures. In our study, SF-12 was used as a single overall composite quality-of-life score for within-sample comparisons, consistent with prior published work that examined health anxiety and QoL among medical students using the SHAI and SF-12 together and reported QoL as a single aggregate score. We have now revised the Methods section to clearly describe the scoring approach applied in this study and have revised the Results to explicitly state that the reported QoL values represent a non–norm-based composite score. We also added a limitation noting that this approach limits direct comparability with studies reporting standard norm-based PCS/MCS values.

Minor comments

Regarding the response to the first review (Comment 2): While the authors have added a discussion regarding students' economic status and university facilities to explain the disparity between private and public universities, these factors were not actually measured in this study and remain speculative. It is recommended to explicitly mention in the Limitations section that these potential contributors were not assessed.

Response:

We thank the reviewer for this important clarification. We agree that factors such as students’ economic status and institutional facility differences were not directly measured in this study and therefore should not be interpreted as confirmed explanatory variables. To address this, we revised the Discussion to clearly frame these points as plausible hypotheses rather than established conclusions, and we added an explicit statement in the Limitations section indicating that financial stress indicators and institutional-level support service measures were not assessed and should be evaluated in future research.

---

## [Decision Letter · Decision Letter 3]

22 Mar 2026

Dear Dr. Al-bawah,

Thank you for submitting your manuscript to PLOS ONE. After careful consideration, we feel that it has merit but does not fully meet PLOS ONE’s publication criteria as it currently stands. Therefore, we invite you to submit a revised version of the manuscript that addresses the points raised during the review process.

We look forward to receiving your revised manuscript.

Kind regards,

Ahmed Abdelwahab Ibrahim El-Sayed,

Academic Editor

PLOS One

**Journal Requirements:**

**Additional Editor Comments:**

Dear Author,

Thank you for your revision. I have finished my review to your paper. The reviewer recommended some certain points that need to be addressed carefully before your submission can be considered further.

Reviewers' comments:

Reviewer's Responses to Questions

**Comments to the Author**

Reviewer #1: All comments have been addressed

Reviewer #5: All comments have been addressed

2. Is the manuscript technically sound, and do the data support the conclusions?

Reviewer #1: Yes

Reviewer #5: Yes

3. Has the statistical analysis been performed appropriately and rigorously?

Reviewer #1: Yes

Reviewer #5: Yes

4. Have the authors made all data underlying the findings in their manuscript fully available?

Reviewer #1: Yes

Reviewer #5: Yes

5. Is the manuscript presented in an intelligible fashion and written in standard English?

Reviewer #1: Yes

Reviewer #5: Yes

Reviewer #1: Here is a concise version for the submission box:

Thank you for the authors' efforts across multiple revisions. The manuscript has improved substantially. However, several issues require clarification before acceptance:

1. Abnormal prevalence of past psychiatric history. Table 1 shows 450/650 participants (69.2%) reporting a personal psychiatric history, yet Figure 2 shows specific disorders at much lower rates (e.g., depression 16.3%). This discrepancy is unexplained and raises concerns about how this variable was defined and measured. Given that it yields an AOR of 3.74, its validity directly affects the regression model's interpretation.

2. Inconsistent reporting of academic dissatisfaction as a predictor. The Abstract and Discussion describe academic dissatisfaction as a significant predictor of health anxiety; however, Table 1 shows p = 0.081 (non-significant), and it does not appear in the multivariable model (Table 3). This overclaiming must be corrected.

3. Misleading figure in the Abstract. The statement "higher prevalence was observed among females (49%)" appears to reflect the proportion of females among HA cases, not the prevalence of HA among females (which is 36.3% per Table 1). These are fundamentally different quantities and should be clearly distinguished.

4. Missing Year 4 in Table 3. The multivariable regression includes Years 2, 3, 5, and 6 but omits Year 4 without explanation. Authors should clarify whether this was intentional and why.

5. Figure 1 p-value notation. "p = 0.000" should be corrected to "p < 0.001" per standard statistical reporting conventions.

Reviewer #5: (No Response)

.

Reviewer #1: No

Reviewer #5: No

---

## [Author Response · Author response to Decision Letter 4]

24 Mar 2026

Response to reviewer #1

We sincerely thank the reviewer for their thoughtful and constructive comments. We appreciate the positive feedback regarding the improvements in the manuscript. We have carefully addressed all concerns, and the manuscript has been revised accordingly. Our detailed responses are provided below:

Comment 1:

Abnormal prevalence of past psychiatric history. Table 1 shows 450/650 participants (69.2%) reporting a personal psychiatric history, yet Figure 2 shows specific disorders at much lower rates (e.g., depression 16.3%). This discrepancy is unexplained and raises concerns about how this variable was defined and measured. Given that it yields an AOR of 3.74, its validity directly affects the regression model's interpretation.

Response:

We thank the reviewer for highlighting this important issue. The apparent discrepancy is due to differences in how these variables were collected. “Past psychiatric history” was assessed using a mandatory binary (yes/no) item, capturing any prior psychological or psychiatric condition, including both formally diagnosed and self-perceived conditions.

In contrast, the reporting of specific psychiatric conditions (Figure 2) was based on a non-mandatory, multiple-response question, allowing participants to optionally select specific disorders. Consequently, not all participants who reported a psychiatric history provided details on specific conditions, leading to lower frequencies for individual disorders.

To clarify this, we have:

• Added a detailed explanation in the Methods section describing how both variables were measured

• Included clarification in the Results section explaining the discrepancy

• Added a limitation in the Discussion acknowledging potential underreporting of specific conditions

Comment 2:

Inconsistent reporting of academic dissatisfaction as a predictor. The Abstract and Discussion describe academic dissatisfaction as a significant predictor of health anxiety; however, Table 1 shows p = 0.081 (non-significant), and it does not appear in the multivariable model (Table 3). This overclaiming must be corrected.

Response:

We appreciate the reviewer’s careful observation. We agree that this was an overstatement.

We have revised the manuscript to ensure consistency:

• Removed academic dissatisfaction as a significant predictor from the Abstract

• Revised the Discussion to clarify that the association did not reach statistical significance and was not included in the multivariable model

• Reframed the interpretation to describe it as a non-significant trend rather than an independent predictor

Comment 3:

Misleading figure in the Abstract. The statement "higher prevalence was observed among females (49%)" appears to reflect the proportion of females among HA cases, not the prevalence of HA among females (which is 36.3% per Table 1). These are fundamentally different quantities and should be clearly distinguished.

Response:

We thank the reviewer for this important clarification. We agree that the previously reported value (49%) reflected the proportion of females among health anxiety cases, rather than the prevalence of health anxiety among females.

We have corrected this in the Abstract to accurately report prevalence:

• Health anxiety prevalence is now stated as 36.3% among females vs. 25.8% among males, consistent with Table 1.

Comment 4:

Missing Year 4 in Table 3. The multivariable regression includes Years 2, 3, 5, and 6 but omits Year 4 without explanation. Authors should clarify whether this was intentional and why.

Response:

We appreciate the reviewer for noting this lack of clarity. Year 4 was indeed included in the regression model but did not show a statistically significant association.

To address this, we have:

• Added a footnote to Table 3 stating:

“Year 1 was used as the reference category. All academic years were included in the model; Year 4 is not shown as it was not statistically significant.”

• Included a brief clarification in the Results section

Comment 5:

Figure 1 p-value notation. "p = 0.000" should be corrected to "p < 0.001" per standard statistical reporting conventions.

Response:

We thank the reviewer for this important correction. In response, we have removed the p-value notation (“p = 0.000”) from within the figure itself and instead reported it correctly in the figure footnote as “p < 0.001”, in accordance with standard statistical reporting conventions. This change improves both clarity and adherence to reporting guidelines.

---

## [Editor Report · Decision Letter 4]

1 Apr 2026

Prevalence and predictors of clinically significant health anxiety among Saudi medical students: a multi-university cross-sectional study

PONE-D-25-43759R4

Dear Author,

We’re pleased to inform you that your manuscript has been judged scientifically suitable for publication and will be formally accepted for publication once it meets all outstanding technical requirements.

Kind regards,

Ahmed Abdelwahab Ibrahim El-Sayed

Academic Editor

PLOS One

l Editor Comments

Dear Authors,

Thank you for your efforts to enhance your study. I can accept your manuscript for publication at PLOS ONE in its current form. Congratulations!

---

## [Editor Report · Acceptance letter]

PONE-D-25-43759R4

PLOS One

Dear Dr. Al-bawah,

I'm pleased to inform you that your manuscript has been deemed suitable for publication in PLOS One. Congratulations! Your manuscript is now being handed over to our production team.

Kind regards,

on behalf of

Dr. Ahmed Abdelwahab Ibrahim El-Sayed

Academic Editor

PLOS One